# Sulforaphane and Benzyl Isothiocyanate Suppress Cell Proliferation and Trigger Cell Cycle Arrest, Autophagy, and Apoptosis in Human AML Cell Line

**DOI:** 10.3390/ijms252413511

**Published:** 2024-12-17

**Authors:** Anna Bertova, Szilvia Kontar, Martina Ksinanova, Alberto Yoldi Vergara, Zdena Sulova, Albert Breier, Denisa Imrichova

**Affiliations:** 1Institute of Molecular Physiology and Genetics, Centre of Biosciences, Slovak Academy of Sciences, Dúbravská Cesta 9, 840 05 Bratislava, Slovakia; anna.bertova@savba.sk (A.B.); szilvia.kontar@savba.sk (S.K.); martina.ksinanova@savba.sk (M.K.); umfgyolv@savba.sk (A.Y.V.); zdena.sulova@savba.sk (Z.S.); 2Institute of Biochemistry and Microbiology, Faculty of Chemical and Food Technology, Slovak University of Technology in Bratislava, Radlinského 9, 812 37 Bratislava, Slovakia

**Keywords:** sulforaphane (SFN), benzyl isothiocyanate (BITC), multidrug resistance, ABCB1 transporter, cyclins, cyclin-dependent kinases, apoptotic proteins, natural product, leukemia

## Abstract

Isothiocyanates (ITCs) are naturally occurring sulfur-containing compounds with diverse biological effects. This study investigated the effects of sulforaphane (SFN, an aliphatic ITC) and benzyl isothiocyanate (BITC, an aromatic ITC) on human acute myeloid leukemia SKM-1 cells, focusing on cell proliferation, cell death, and drug resistance. Both drug-sensitive SKM-1 cells and their drug-resistant SKM/VCR variant, which overexpresses the drug transporter P-glycoprotein, were used. SFN and BITC reduced cell viability in a dose-dependent manner, with BITC showing greater potency. IC50 values ranged from 7.0–8.0 µM for SFN and 4.0–5.0 µM for BITC in both cell types, with only slight differences between the variants. Both ITCs induced autophagy as evidenced by increased LC3-II production and caused a significant increase in the sub-G0/G1 cell population, especially with BITC. Apoptosis was more pronounced after BITC treatment, whereas SFN had a weaker effect. These results suggest that autophagy may act as a defense mechanism in response to ITC-induced apoptosis in human AML cells.

## 1. Introduction

Acute myeloid leukemia (AML) has a heterogeneous etiology resulting in diverse pathophysiologic manifestations. The main features of this aggressive disease include rapid uncontrolled proliferation of malignant myeloid cells leading to massive expansion and infiltration of poorly differentiated blasts in the bone marrow and peripheral blood, replacing normal healthy white blood cells. With a characteristically higher incidence in older adults aged 60–65 years, AML is the most common form of acute leukemia [1]. In newly diagnosed patients, standard chemotherapy leads to complete remission in 60–80% of cases, but with repeated cycles of chemotherapy, relapse occurs in approximately 30% of younger patients and 70–80% of older patients [2]. This has led to the need to expand therapeutic options, and new therapeutic agents have already been approved for use alone or in combination with conventional chemotherapeutic agents [3,4]. However, even with the latest therapies, the development of drug resistance is a major obstacle in the treatment of AML [5,6]. In up to one-third of patients, this is due to overexpression of the ABCB1 gene encoding P-glycoprotein (P-gp), which can lead to multiple drug resistance (MDR), a specific phenotype of neoplastic cells [7]. P-gp is an ATP-dependent membrane pump that is physiologically expressed in excretory tissues, including kidney, liver and others, as well as in barrier structures in the placenta and blood–brain barrier, where it is involved in the transport and excretion of endogenous compounds and xenobiotics [8]. Many active small molecules (such as isothiocyanates) that occur naturally in our diet have the potential to serve as new drugs or adjuvants in the treatment of AML. Although the anti-cancer and health-promoting effects of natural and synthetic isothiocyanates (ITCs) have been known for more than five decades [9], their mechanisms of action are still the subject of intense research [10]. Many biological activities, such as anti-inflammatory, antioxidant, neuroprotective, and anti-tumor, have been described for ITCs, which are mainly found in vegetables of the *Brassicaceae* family. They are hidden in these plants in the form of glucosinolates, from which they can be released by hydrolysis by the enzyme myrosinase [11]. Both glucosinolates and ITCs are versatile organosulfur compounds with universal β-thioglucoside-N-hydroxysulphate and ITC groups, respectively. These are complemented by a variety of aliphatic or aromatic side branches. Natural ITCs exhibit the electrophilic reactivity of the ITC group with -SH, -NH_2_, and -OH groups in biomolecules [9,12]. The side chains of ITCs are responsible for the physico-chemical properties of the molecule, particularly the lipophilicity, which influences its ability to penetrate cells and may influence the intensity of its effect on the target structure [13].

Chemopreventive and therapeutic activities have been demonstrated, namely the ability to suppress tumor cell proliferation and induce apoptosis in cell lines as well as in animal models [14]. This has mainly been done in studies using solid tumor cells and animal models, but less is known about their antileukemic effects, which remain poorly understood [15]. In the present work, we focused on the effects of two natural ITCs: aromatic benzyl isothiocyanate (BITC, mainly found in broccoli, cabbage, and watercress) and sulforaphane (SFN, mainly found in broccoli, cabbage, cauliflower, and kale), on P-gp-negative SKM-1 human AML cells and their P-gp-positive variants, SKM/VCR. We evaluated their effect on cell proliferation, cell cycle progression, and the induction of apoptosis in relation to the expression levels of proteins involved in the regulation of these processes.

## 2. Results

### 2.1. Effects of SFN and BITC on Cell Viability at Different Concentrations

We first examined the effects of two structurally different ITCs (Figure 1) on cell viability using SFN or BITC (0–16 μM) over 24 and 48 h periods. Cell counting, viability, and membrane integrity were assessed using the CASY model TT cell counter. Both ITCs showed a monotonic dose-dependent effect, with a decrease in cell viability compared to controls (Figure 2A). The data indicated that BITC was a more potent inhibitor than SFN in both cell lines.

The cell dehydrogenase-dependent reduction of tetrazolium salt to formazan (MTS assay) was used to assess changes in metabolic activity induced by cell death processes, as shown in Figure 2B. BITC showed a significantly higher inhibitory effect compared to SFN, with SFN having a higher IC_50_ in both cell lines (SKM-1: 7.31 μM; SKM/VCR: 7.93 μM), whereas BITC had a lower IC_50_ (SKM-1: 4.15 μM; SKM/VCR: 4.76 μM) (Figure 2C). In the SKM-1 cell line, an increase in metabolic activity was observed at low concentrations of SFN (2 and 4 μM). A similar but weaker effect was observed in the SKM/VCR cell line. However, these increases were not associated with an increase in cell number, as confirmed by direct cell counting (Figure 2A). Similar behavior was observed when SFN was applied to P-gp-negative and P-gp-positive murine lymphoblastic leukemia L1210 cells [13]. In contrast, BITC, even at low concentrations, did not induce such an unusual discrepancy between cell number and metabolic activity.

The calculated resistance index (RI) was 1.08 for SFN and 1.15 for BITC, not significantly different from 1. It can be concluded that P-gp does not confer significant resistance to these agents (Figure 2C).

### 2.2. Apoptosis Induced by SFN and BITC Treatment in SKM-1 and SKM/VCR Cells

The apoptosis-inducing effects of SFN and BITC were investigated by flow cytometry on SKM-1 and SKM/VCR cells stained with annexin V and propidium iodide (PI). Annexin V is a calcium-dependent protein with high affinity for phosphatidylserine that translocates to the outer surface of the plasma membrane during apoptosis. PI stains double-stranded DNA but cannot penetrate intact plasma membranes, so its uptake indicates necrosis in the cells. The results are shown in Figure 3. The lower left quadrant represents viable cells, the lower right quadrant represents early apoptotic cells, the upper left quadrant represents necrotic cells, and the upper right quadrant represents late apoptotic cells.

Treatment of cells with 8 μM SFN for 8 h induced negligible apoptosis compared to 8 μM BITC, which significantly increased the proportion of annexin V-positive apoptotic cells (sum of cells in both right quadrants). Specifically, in control SKM-1 cells, the proportion increased from 8.2% to 24.9%, while in SKM/VCR cells, it rose from 4.3% to 12.8% (Figure 3A). Prolonged exposure to 8 μM SFN (12 h) induced apoptosis only in the SKM-1 cell line, with annexin V-positive cells increasing from 5% to 8.7%. In P-gp-positive SKM/VCR cells, the increase was minimal. In contrast, treatment with 8 μM BITC had a much stronger apoptotic effect. In SKM-1 cells, BITC increased the proportion of apoptotic cells from 5% to 36.6%, while in SKM/VCR cells, the increase was from 5.2% to 27.7% (Figure 3B). A time-dependent induction of apoptosis was confirmed after 24 h of incubation with both tested ITCs. A significant increase in apoptotic cells was observed in both cell lines; however, the P-gp-positive SKM/VCR cell line demonstrated significantly greater resistance to ITCs compared to the P-gp-negative SKM-1 cell line. Specifically, treatment with 8 μM SFN increased the proportion of apoptotic cells from 5.9% to 19.6% in SKM-1 and from 4.6% to 10.8% in SKM/VCR. BITC proved to be a much stronger inducer of apoptosis, raising the proportion of apoptotic cells from 5.9% to 53.8% in SKM-1 and from 4.6% to 41.5% in SKM/VCR cells (Figure 3C).

Flow cytometric assay showed that both ITCs induced apoptosis in SKM-1 and SKM/VCR cell lines, with P-gp-deficient cells being more sensitive to these compounds. We further investigated the potential mechanisms by which SFN and BITC influence apoptosis in the AML cell lines tested. Cells were treated with SFN and BITC (0, 4, and 8 μM) for 12 h, and the levels of several apoptosis-regulating proteins were examined by Western blotting. In Western blotting, it is important to find a suitable internal control. Unfortunately, our experiments showed changes in the levels of commonly used housekeepers such as GAPDH and β-actin under the influence of SFN or BITC (Appendix A). Because of these issues, we used red staining with Ponceau S to control the loading of proteins in the different electrophoretic lines and the efficiency of electroblotting.

As shown in Figure 4, increasing doses of SFN significantly induced the pro-apoptotic protein Bax in both SKM-1 and SKM/VCR cells. However, SFN also significantly increased the levels of the anti-apoptotic proteins Bcl-2 and Mcl-1 in both cell lines. BITC treatment also significantly increased the levels of Bcl-2 and Mcl-1 but surprisingly caused a significant decrease in the levels of Bax protein in SKM-1 cells. This effect was not observed in the P-gp-positive SKM/VCR cell line.

Western blot analysis also showed that SFN significantly upregulated caspase 8 and caspase 9 levels, although there were no significant changes in caspase 3 levels in either cell line. Conversely, BITC significantly downregulated caspase 3 levels in both cell lines. However, BITC induced caspase 8 expression in the P-gp-negative SKM-1 cell line and caspase 9 expression in the P-gp-positive SKM/VCR cell line.

Taken together, these data suggest that BITC differentially affects the protein levels of Bax, caspase 8, and caspase 9 in SKM-1 cells compared to SKM/VCR cells (Figure 4). The most striking differences between the effects of these two ITCs were observed in the regulation of caspase 3 and Bax.

### 2.3. SFN and BITC Treatment Induced Conversion of Autophagy Markers LC3-I to LC3-II

Since relatively low levels of dead cells were observed in both SKM-1 and SKM/VCR cells during flow cytometry with annexin V and PI staining, particularly under the influence of SFN, we questioned whether an alternative mechanism might contribute to cell damage by SFN and BITC. We therefore investigated the possible involvement of autophagy by analyzing the protein levels of its markers, microtubule-associated protein 1 light chain 3 (LC3), in particular its 18 kDa LC3-I and 16 kDa LC3-II proteolytic fragments. The conversion of cytosolic LC3-I to membrane-bound LC3-II is a well-established marker of autophagic activity, although it does not necessarily indicate completion of the process [16]. To assess this, we performed Western blotting to monitor the levels of LC3-I and LC3-II (Figure 5). The results showed a dose-dependent increase in the conversion of LC3-I to LC3-II following treatment with SFN and BITC (0, 4, and 8 μM) for 12 h in both cell lines. Notably, BITC was also more effective than SFN in inducing autophagy, as indicated by a more pronounced conversion of LC3-I to LC3-II at equivalent concentrations. These results suggest that both SFN and particularly BITC have significant potential to induce autophagy as part of the cellular response to ITC-induced chemical stress.

### 2.4. Cell Cycle Alterations Induced by SFN and BITC in SKM-1 and SKM/VCR Cells

To explore the mechanism underlying the antiproliferative effects of the two ITCs studied, we measured the cell cycle distribution at different phases. Flow cytometric analysis revealed that treatment with SFN at a concentration of 12 μM for 12 h caused a significant decrease in the percentage of cells in the G0/G1 phase, from 35.9% to 24.2% (* *p* < 0.05) in parental SKM-1 cells (Figure 6A). A more modest reduction in the G0/G1 phase, from 42.7% to 35.3% (* *p* < 0.05), was observed in the SKM/VCR variant, accompanied by a significant increase in the proportion of cells in the G2/M phase, from 21% to 28% (* *p* < 0.05) (Figure 6B). Under these conditions, SFN did not induce significant changes in the proportion of cells in S phase.

In comparison, BITC treatment had a stronger effect on cell cycle dynamics. Significant changes in cell cycle distribution were observed after only 4 h of treatment with 10 μM BITC. In SKM-1 cells, BITC significantly increased the percentage of cells in the sub-G1 phase (indicating fragmented or degraded DNA) from 0.7% to 6.2% (** *p* < 0.01, Figure 6C). In SKM/VCR cells, the increase was even more pronounced, from 1.1% to 11% (** *p* < 0.01, Figure 6D). BITC also significantly reduced the proportion of cells in the G0/G1 phase, from 36% to 22.5% (** *p* < 0.01) in SKM-1 cells and from 47.6% to 25% (** *p* < 0.01) in SKM/VCR cells. In SKM-1 cells, this reduction in G0/G1 phase was accompanied by a slight increase in the proportion of cells in S and G2/M phases. In contrast, SKM/VCR cells showed a more definitive response with a significant cell cycle arrest in the S phase, as indicated by an increase in the proportion of cells in this phase from 28.5% to 42.4% (** *p* < 0.01).

To understand the cell cycle changes induced by SFN and BITC, we analyzed the protein levels of key cell cycle regulators using Western blotting. We focused on cyclin-dependent kinase 1 (Cdk1), which works with cyclins A and B1 to regulate the G2/M transition and its completion, and cyclin-dependent kinase 2 (Cdk2), which works with cyclins E and A to regulate the G1/S transition and its progression (Figure 7).

Interestingly, Cdk2, cyclin B1, and cyclin D levels were lower in SKM/VCR cells compared to their sensitive counterparts. Both ITCs increased Cdk2 levels in SKM/VCR cells in a concentration-dependent manner. However, BITC decreased Cdk2 levels in SKM-1 cells in a concentration-dependent manner. In contrast, Cdk1 levels showed no significant differences between SKM-1 and SKM/VCR cells, nor did they change significantly after ITC treatment.

While SFN only slightly reduced the levels of cyclin D and cyclin E in SKM-1 cells, especially at the highest concentration, BITC appeared to be more effective, additionally reducing the levels of cyclins A and B1. The reduced levels of cyclin B1 and D in SKM/VCR cells were not significantly affected by either ITC, except for the overexpression of cyclin B1 at the highest SFN concentration.

In SKM/VCR cells, SFN treatment resulted in a concentration-dependent increase in cyclin A and E levels, whereas BITC treatment resulted in a decrease. The exact mechanism underlying the cell cycle changes observed in Figure 6 cannot yet be fully explained by the changes in cell cycle checkpoint protein levels (Figure 7). However, it is clear that changes in the levels of cyclin-dependent kinases and cyclins are closely related to the observed changes in the cell cycle. It is also important to note that SFN and BITC induce different changes in the cellular levels of these proteins and that the effects of ITCs differ between SKM-1 and SKM/VCR cells.

## 3. Discussion

Despite significant advances in the chemotherapy of hematological malignancies, which have enabled the treatment of previously incurable cases, the development of resistance in leukemia cells remains a major obstacle to effective therapy [5]. As a result, several research teams are focusing on the discovery of natural compounds or their synthetic analogs that can overcome cellular drug resistance. ITCs released from plant glucosinolates, mainly from the *Brassicaceae* family, through the myrosinase reaction, have emerged as some of the most promising candidates, as demonstrated by extensive research [17,18]. These ITCs often exhibit significant anticancer potential, inhibiting cancer cell growth in both in vitro and in vivo experiments by inducing apoptosis [19,20]. A number of scientific studies have focused mainly on the main members of the ITC family that are commonly found in the human diet [10], including SFN (mainly found in cauliflower, cabbage, broccoli, and turnip), phenethyl isothiocyanate and BITC (both mainly found in broccoli, watercress, and cabbage), and allyl isothiocyanate (mainly found in horseradish, mustard, radish, and wasabi). The main tasks in the molecular research on the anticancer activities of natural ITCs are to determine whether they exhibit specificity against different types of malignancies and to elucidate the mechanisms involved [21]. A key feature of these bioactive molecules, with respect to their anticancer potential, is their selective cytotoxic and pro-apoptotic effects, which are predominantly directed against neoplastic transformed cells, originating from both solid tumors and onco-hematological malignancies [17,22].

In this study, we investigated the aliphatic SFN and the aromatic BITC (Figure 1) as potential inhibitors of leukemia cell proliferation and inducers of cell death. We used these two isothiocyanates as typical examples of aliphatic and aromatic isothiocyanates commonly found in plants. Both contain the reactive group -NCS, which in terms of reactivity in both isothiocyanates shows approximately the same affinity for nucleophilic groups in the order -SH > -NH_2_ > -OH [8]. The aliphatic or aromatic skeleton attached to this group modulates the overall physicochemical properties (lipophilicity, molecular volume, etc.) responsible for the distribution of the substance between the environment and the cell, as well as within the cell [12]. In addition, we evaluated whether the presence of P-gp, a key molecular factor in the MDR phenotype, influences the cytotoxic efficacy of these ITCs.

Our results show that treatment with SFN and BITC significantly reduced the viability of both SKM-1 and SKM/VCR cell variants in a dose-dependent manner, as determined by direct cell counting (Figure 2A) and MTS assay (Figure 2B). SFN was less potent than BITC, with IC_50_ values of 7.31 µM in SKM-1 cells and 7.93 µM in SKM/VCR cells, compared to BITC values of 4.15 µM in SKM-1 cells and 4.76 µM in SKM/VCR cells after 24 h (Figure 2C). We have also validated the measurements on SKM-1 cells and their P-gp-positive variants in another AML cell line, MOLM-13, with very similar effects. These findings are consistent with previously reported inhibitory concentrations in other leukemia cell models, where SFN exhibited IC_50_ values in the range of 1–7 µM [23]. Most importantly, the presence of the P-gp phenotype only slightly reduced the sensitivity to both ITCs tested. A unique property of SFN is that at lower concentrations it leads to an increase in the MTS signal without a corresponding increase in cell number as determined by direct counting (Figure 2A,B). This phenomenon suggests metabolic activation, specifically related to increased activity of reduced nicotinamide dinucleotides and enhanced dehydrogenase enzyme systems [24]. This effect has been reported several times for SFN [13,25,26].

SFN and BITC have been reported to have cytotoxic effects across various cancer cell types by inhibiting growth and inducing apoptotic cell death, including cells derived from diverse hematological malignancies [14,23,27,28]. Our results suggest that SFN has only a modest apoptosis-inducing effect on the parental SKM-1 cell line and even less on its P-gp-positive counterpart. In previous work with murine L1210 lymphoblastic leukemia cells, we found that the predominant cellular response to SFN was entry into autophagy, and the contribution of apoptosis appeared to be modest [13,29]. It is reasonable to assume that apoptosis and autophagy are potential cellular responses following the introduction of a toxic agent, with the former leading definitively to cell death, whereas the latter may either facilitate repair and survival or alternatively lead to cell death. Consistently, as in our cells in human prostate cancer cells PC-3 and LNCaP, SFN-triggered autophagy was linked to an increased formation of autophagosomes, which corresponded to elevated levels of 18 and 16 kDa proteolytic fragments of the LC3 protein [30]. When cells were treated with a specific autophagy inhibitor (3-methyladenine), LC3 localization to autophagosomes was reduced, but this led to increased cytosolic release of cytochrome c, resulting in apoptotic cell death. SFN and BITC are compounds known not only to induce apoptotic cell death but also to promote autophagy [13,31]. Several authors have suggested that there is a switch between autophagy and apoptosis when SFN is used [32,33]. The fate of cells in autophagy depends on the ability of reparative processes to bring the cells to a viable state where they can continue to proliferate, or, if unsuccessful, the cells will enter death processes [34]. Recent studies have shown that ITC can induce autophagy by upregulating autophagy-related proteins, in particular LC3, which is recruited to autophagosomes, thereby exerting a protective effect on cancer cells [35,36]. Several anticancer agents have been shown to induce autophagy, which may either confer protective effects against drug-induced cytotoxicity or increase sensitivity to therapeutic agents [37]. Previous studies from our laboratory have shown that SFN not only induces apoptosis in a murine leukemia cell line but also promotes autophagy, as detected by WB of LC3 protein fragments and confocal microscopy with monodansyl cadaverine [13,29]. To further elucidate the autophagic process, we examined the processing of the LC3-II protein, which is characteristic of the activation of the autophagic pathway. Our findings demonstrate that both SFN and BITC mediate autophagy in human leukemic cell lines SKM-1 and SKM/VCR, evidenced by the increased formation of the 16 kDa LC3-II proteolytic product, which is a reliable marker of autophagy (Figure 5).

On the other hand, BITC appeared to have a more potent apoptotic effect, showing a significant concentration-dependent increase in apoptosis in both SKM-1 and SKM/VCR cell lines (Figure 3). These results suggest that BITC is a more potent agent than SFN in inducing apoptosis, even in multidrug-resistant cells. Thus, P-gp does not antagonize the cytotoxic effect of BITC. However, BITC appears to be able to inhibit the transport activity of P-gp [38], which is another activity that may be relevant.

The change in the pro/anti-apoptotic ratio led to the activation of mitochondrial dysfunction, caspase activation and apoptosis. To further elucidate the mechanisms underlying the SFN- and BITC-induced suppression of cell viability, we evaluated the expression levels of Bcl-2 family proteins by Western blot analysis. Notably, both SFN and BITC treatment resulted in a significant, dose-dependent increase in the levels of anti-apoptotic proteins Bcl-2 and Mcl-1 in both cell lines. However, the expression of the pro-apoptotic protein Bax was significantly increased in SKM-1 cells after SFN treatment. In contrast, BITC downregulated Bax expression in this parental cell line (Figure 4, Appendix A). Induction of programmed cell death (apoptosis) is regulated by caspases, which activate two distinct apoptotic pathways: the intrinsic mitochondrial pathway involving caspase 9 and the extrinsic death receptor pathway associated with activation of caspase 8. Based on our results, we hypothesize that both SFN and BITC can induce caspase-dependent apoptosis by stimulating the activation of both caspase 9 and caspase 8 in a concentration-dependent manner in human AML cell lines. Importantly, this apoptotic process appears to be influenced by the specific phenotype of the cell line. Apoptosis is observed in cells following treatment with BITC and, to a lesser extent, SFN, as shown in Figure 3. The significant progression of apoptosis is prevented by the overexpression of two anti-apoptotic proteins, Bcl-2 and Mcl-1 (Figure 5). Although a slight overexpression of initiator caspases (8 and 9) is observed after treatment, the expression of the executioner caspase appears to be suppressed. Furthermore, there was no evidence of proteolytic fragments (an indication of their activation) of either of these two caspases. Consistent with previous work on murine leukemia L1210 cells [13,29], the contribution of apoptosis was relatively low when SFN was used. The situation was different when the more aromatic, lipophilic BITC was used, as we observed a higher proportion of cells undergoing apoptosis (Figure 3). However, even in this case, apoptosis could not be considered the predominant mode of cell death. These results were obtained when cells were simply passaged in media containing BITC or SFN. Cells can repair damaged functions and prolong proliferation through autophagic processes in a single passage. In a previous study [29], we demonstrated that repeated passage of murine leukemia L1210 cells in the presence of sublethal concentrations of SFN reduced cell viability. Therefore, we hypothesize that prolonged exposure to ITCs over multiple passages will increase their cytotoxicity, even at sublethal concentrations.

In addition, the effect of the tested ITCs on cell cycle progression showed a significant increase in the proportion of cells in the sub-G0/G1 phase, particularly after BITC treatment, with SFN showing a similar effect, although to a lesser extent (Figure 6C,D). This increase in sub-G0/G1 cells was associated with a marked downregulation of caspase 3 and an overexpression of Bcl-2 (Figure 4). This suggests that autophagy may be activated as a cellular defense mechanism in response to SFN- and BITC-induced apoptosis in human AML cells. We propose that the enhanced pro-apoptotic potential of BITC compared to SFN may contribute to a more pronounced activation of autophagy, serving as a pro-survival signal in these malignant cells. Our findings are consistent with other studies that have reported BITC and PEITC as activators of survival autophagy pathways in advanced prostate and human breast cancer cells, which may facilitate resistance to treatment with these compounds [31,39].

SFN and BITC have been shown to suppress the proliferation of various cancer cells in vitro by inducing cell cycle arrest [28,40]. Flow cytometry analysis (Figure 6) showed that SFN effectively reduced the accumulation of cells in the G0/G1 phase and induced cell cycle arrest in the G2/M phase in both cell lines tested. Notably, G2/M arrest was also observed in oral, bladder, and prostate cancer cell lines, which showed a similar pattern of G2/M arrest accompanied by an increase in the sub-G1 cell population [41,42,43]. The differential effects of BITC on the human leukemia cells tested were interesting. Treatment with this aromatic ITC led to a significant increase in the sub-G1 population, an indicator of apoptosis, in both cell lines. In addition, BITC treatment led to cell cycle arrest in the S phase, which was associated with a significant downregulation of cyclin A and cyclin-dependent kinase 1 (Cdk1) (Figure 7). This downregulation corresponded to a decrease in the proportion of cells in the G0/G1 phase, and this effect was more pronounced in the P-gp-positive SKM/VCR cell line. We conducted these experiments at various times and concentrations of ITCs. The data presented in Figure 6, obtained after 12 h of incubation with 4, 8, and 12 μM, yielded relevant and reproducible results. At higher concentrations and longer incubation times, a significant proportion of the cells were in the sub-G1 phase, and the remaining live cells were markedly reduced in number, preventing reliable determination of the cell cycle through flow cytometry.

These findings underscore the potential of BITC as a promising therapeutic candidate in overcoming drug resistance in acute myeloid leukemia. Further investigations are warranted to explore the underlying mechanisms of action and the clinical implications of these compounds in treatment strategies for hematological malignancies. Further investigation of ITCs in cancer therapy may pave the way for innovative approaches to improve treatment efficacy and patient outcomes.

## 4. Materials and Methods

### 4.1. Chemicals

SFN [1-isothiocyanato-4-(methylsulfinyl)butane] and BITC [(isothiocyanatomethyl)benzene] were purchased from Sigma-Aldrich (Merck Life Science, Bratislava, Slovakia). Chemicals were of analytical grade.

### 4.2. Cell Culture and Culture Conditions

In this study, an acute myeloid leukemia cell line SKM-1 (ACC 547) derived from the peripheral blood of a 76-year-old man with acute myeloid leukemia (AML M5) that had progressed to myelodysplastic syndrome (MDS) was used as an experimental model. The SKM-1 cell line was obtained from the Leibniz-Institut-Deutsche Sammlung von Mikroorganismen und Zellkulturen GmbH (Braunschweig, Germany). A drug-resistant SKM/VCR cell line overexpressing the drug transporter ABCB1 was established by long-term culture with the anticancer drug vincristine (VCR; Sigma-Aldrich, St. Louis, MO, USA). VCR concentrations were gradually increased to a final concentration of 60 nM, as previously described [44]. These cells express high levels of the ABCB1 gene at both the mRNA and protein levels and exhibit P-gp efflux activity in a calcein retention assay. Both cell lines were grown in RPMI 1640 medium containing 12% fetal bovine serum, 100,000 units/L penicillin, and 50 mg/L streptomycin (both purchased from Sigma-Aldrich, St. Louis, MO, USA) in a humidified atmosphere with 5% CO_2_ at 37 °C.

### 4.3. Determination of the Number and Viability of Cells

The antiproliferative effect of SFN and BITC on human leukemic SKM-1 and SKM/VCR cell growth was determined by counting the absolute number of viable cells using a CASY Model TT Cell Counter (Roche Applied Sciences, Madison, WI, USA) by measuring plasma membrane integrity according to the manufacturer’s protocol. Briefly, cells (3 × 10^5^ cells/mL) were seeded in 24-well plates (2 mL per well) and incubated with SFN or BITC in the concentration range of 2–16 µM for 24 h and 48 h. Results represent the mean ± SD of five independent experiments.

### 4.4. Cell Viability Using an MTS Assay and Determination of IC_50_ Values

Cellular metabolic activity was assessed using 3-(4,5-dimethylthiazol-2-yl)-5-(3-carboxymethoxyphenyl)-2-(4-sulfophenyl)-2H-tetrazolium inner salt (MTS) according to the manufacturer’s protocol (Promega, Madison, WI, USA). SKM-1 and SKM/VCR cell lines were seeded in 24-well plates at a density of 6 × 10^5^ cells/well and treated with 0, 2, 4, 6, 8, 10, 12, 14, 16, 18, and 20 µM SFN or BITC for 24 h at 37 °C. After cell incubation, 100 µL aliquots of each sample were incubated with 20 µL of MTS assay solution in a 96-well plate for 3 h. The absorbance of each well was read at 490 nm using a µQuant universal microplate spectrophotometer (BioTek Instruments Inc., Winooski, VT, USA). Experiments were performed in quadruplicate, and the average relative absorbance was used as a measure of cell metabolic activity. Drug concentrations that inhibited cell proliferation to 50% of the control (IC_50_) were determined from at least four independent quadruplicate experiments for each treatment. IC_50_ values were calculated according to the Equation (1) by nonlinear regression using SigmaPlot 8.02 software (Systat Software, Inc., San Jose, CA, USA):(1)$$N=a+A\times exp\left[\ln\left(0.5\right)\times {\left(\frac{c}{{IC}_{50}}\right)}^{n}\right]$$
where N in % is the metabolic activity of cells after drug treatment at concentration c; a + A in % is the metabolic activity of control/untreated cells; A represents the metabolic activity suppressed by the respective drug; IC_50_ is the half-maximal inhibitory concentration; and *n* is the order exponent for cytotoxic effects. The data represent computed values ± standard error with 30 degrees of freedom.

The resistance index (RI) of the two cell lines was calculated from the median inhibitory concentration (IC_50_) using the following formula RI = IC_50_ resistant cells (SKM/VCR)/IC_50_ sensitive cells (SKM-1).

### 4.5. Cell Apoptosis Assay

Cell apoptosis was detected using the annexin V-FLUOS staining kit (Roche Diagnostics, Mannheim, Germany) according to the manufacturer’s instructions. Briefly, SKM-1 and SKM/VCR cells (1.25 × 10^6^ cells/well) were plated in 6-well plates with RPMI 1640 medium for overnight growth. After 24 h of growth, cells were treated with SFN or BITC at concentrations of 0, 4, and 8 μM for 8 h, 12 h, and 24 h. After treatment, 1 × 10^6^ cells from each treatment were washed with PBS. The cell pellet was resuspended in 100 μL of annexin V-FLUOS labelling solution/propidium iodide (PI) in the dark. It was incubated for 15 min at room temperature and analyzed on an Accuri C6 flow cytometer (BD Bioscience, San Jose, CA, USA).

### 4.6. Cell Cycle Analysis

Cells were seeded in 6-well plates (1.25 × 10^6^ cells/well) for 24 h. They were then incubated in the presence of SFN or BITC at concentrations of 0, 4, 8, 10, and 12 μM. After 4 h and 12 h of culture, the cells were harvested (2 × 10^6^ cells) and washed with PBS. They were then fixed in cold 70% ethanol at 4 °C for 45 min. After centrifugation and washing with PBS, cells were resuspended in 100 μL RNase buffer containing 1 μg/mL RNase A (Thermo Fischer Scientific, Waltham, MA, USA) and incubated at 37 °C in the dark for 30 min. They were then stained with PI, incubated at 37 °C for 30 min, and analyzed by flow cytometry on an Accuri C6 (BD Bioscience, San Jose, CA, USA).

### 4.7. Western Blotting

SKM-1 and SKM/VCR cells were treated with 0, 4, 8, and 12 µM SFN or BITC and incubated for 12 h. After each treatment, the cells were harvested and lysed with RIPA lysis buffer containing protease inhibitor cocktail (Sigma-Aldrich, St. Louis, MO, USA) to extract total proteins. Samples were heated at 100 °C for 15 min, then the protein samples were separated by SDS-PAGE gel electrophoresis according to the protocol published by Laemmli [45]. Proteins were transferred to nitrocellulose membrane (GE Healthcare Europe GmbH, Vienna, Austria) by electroblotting. After transfer, the membranes were blocked with 5% nonfat dry milk solution or in 5% bovine serum albumin at room temperature for 1 h. Western blotting was performed to detect the expression of BAX (N-20): sc-493, Bcl-2 (C-2): sc-7382, cyclin A (BF683): sc-239, cyclin B1 (GNS1): sc-245, cyclin E (E-4): sc-377100, Mcl-1 (B-6): sc-74436, all from Santa Cruz Biotechnology, Dallas, TX, USA, caspase 3 antibody #9662, CDK2 (78B2), cdc2 antibody #77055, all from Cell Signaling Technology, Danvers, MA, USA, anti-caspase 8 antibody (ab25901), anti-caspase 9 antibody (ab63342), anti-cyclin D1 [EPR2241] C-terminal, all from Abcam, Cambridge, UK, and LC3B antibody (L7543) from Sigma-Aldrich, St. Louis, MO, USA. The membranes were incubated with a specific primary antibody overnight at 4 °C. The membranes were then incubated with the secondary antibody IgG kappa binding protein (m-IgGκ BP) or mouse anti-rabbit IgG-HRP (both purchased from Santa Cruz Biotechnology, Dallas, TX, USA) conjugated to horseradish peroxidase (HRP) for 1 h at room temperature. HRP signals were visualized using an ECL detection system (GE Healthcare Europe GmbH, Vienna, Austria) on an Amersham Imager 600 system (GE Healthcare Europe GmbH, Pittsburgh, PA, USA). Quantitative analysis of each immunoreactive blot was performed to measure the intensity of the band signal using the National Institutes of Health ImageJ 1.52o program.

### 4.8. Ponceau S Staining

Protein samples were separated by SDS-PAGE gel electrophoresis according to the protocol published by Laemmli [45]. Proteins were transferred from the gel to a nitrocellulose membrane (GE Healthcare Europe GmbH, Vienna, Austria) by electroblotting. The membrane was stained for 10 min on a shaker with a solution of Ponceau S, 0.1% Ponceau S (*w*/*v*) in 5% acetic acid (*v*/*v*) (Sigma-Aldrich, St. Louis, MO, USA), at room temperature, which is a suitable reagent for use in electrophoresis studies. After scanning, the membranes were rinsed again in distilled water for 2–3 min until the staining was completely removed, and we proceeded with the blocking and antibody incubation steps. 

### 4.9. Statistical Analysis

Data are expressed as the mean ± SD. Differences were evaluated using one-way ANOVA for multiple comparisons and *t*-test for 2-group comparisons. Each experiment was repeated at least three times. *p* < 0.05 was considered statistically significant.

## 5. Conclusions

The key finding of this study is that both SFN and BITC effectively inhibit the growth of SKM-1 acute myeloid leukemia (AML) cells and their P-glycoprotein (P-gp)-overexpressing drug-resistant variant, SKM/VCR. The more lipophilic, aromatic BITC has an IC_50_ of less than 5 μM, while the more hydrophilic, aliphatic SFN has an IC_50_ of less than 8 μM. Both compounds exhibit significant cytotoxic effects, with BITC demonstrating a greater ability to induce apoptosis, even in multidrug-resistant cells.

SFN primarily induces cell cycle arrest in the G2/M phase, while BITC not only increases the sub-G1 population, indicative of apoptosis, but also downregulates key cell cycle regulators, suggesting its ability to overcome drug resistance mechanisms. The greater lipophilicity of BITC likely enhances its potency, contributing to its more pronounced apoptotic effect compared to SFN.

These differential effects on apoptotic pathways suggest that BITC may be more effective in suppressing the growth of AML cells, including those with P-gp-mediated drug resistance. Therefore, BITC may represent a promising option for targeting resistant AML, providing a strong rationale for further exploration of its potential to overcome treatment-resistant leukemia.

## Figures and Tables

**Figure 1 ijms-25-13511-f001:**
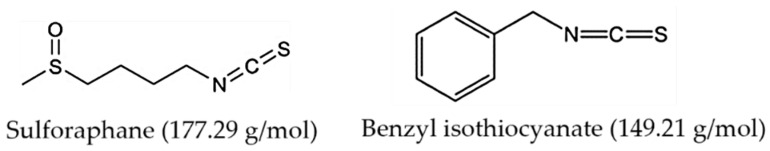
Chemical structures of SFN and BITC.

**Figure 2 ijms-25-13511-f002:**
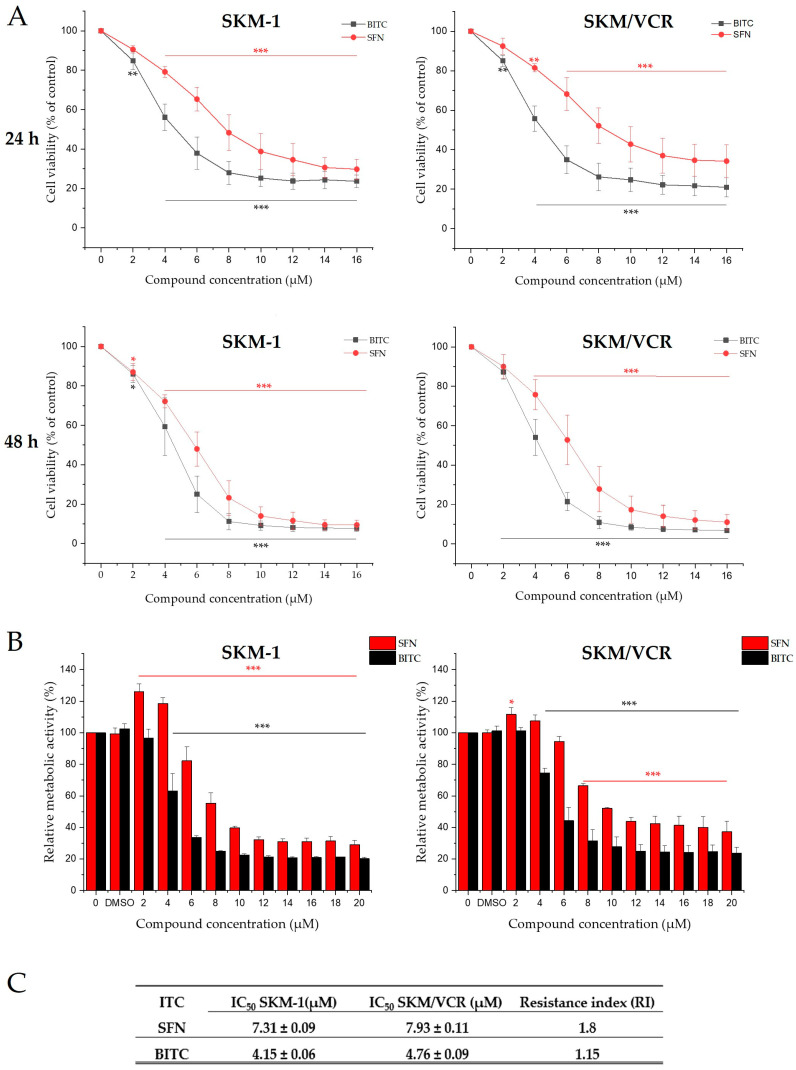
SFN and BITC inhibited the viability of SKM-1 and SKM/VCR cells. (**A**) Dose-dependent inhibition of cell proliferation by ITCs in SKM-1 and SKM/VCR cell lines. Cells were exposed to increasing concentrations of SFN or BITC from 2 µM to 16 µM for 24 h and 48 h. The values shown represent the mean ± SD of five independent experiments. One-way ANOVA with Bonferroni post hoc test was used to assess the significance of differences in the evaluated parameters. (**B**) Inhibitory effect of ITCs on the viability of SKM-1 and SKM/VCR cells. Cells were treated with SFN or BITC (0–20 μM) for 24 h. Cell viability was measured by MTS assay. (**C**) IC_50_ values for SFN or BITC were obtained by nonlinear regression of SKM-1 and SKM/VCR cell proliferation data according to the equation. Results are expressed as the mean ± SD of 4 experiments. One-way ANOVA with Tukey’s post hoc test (* *p* < 0.05, ** *p* < 0.01, *** *p* < 0.001 versus control cells) was used to assess the significance of differences in the evaluated parameters. Resistance indices (RI), representing the ratio of the IC50 values of the resistant cell variants (SKM/VCR) to their sensitive counterparts (SKM-1), are also shown.

**Figure 3 ijms-25-13511-f003:**
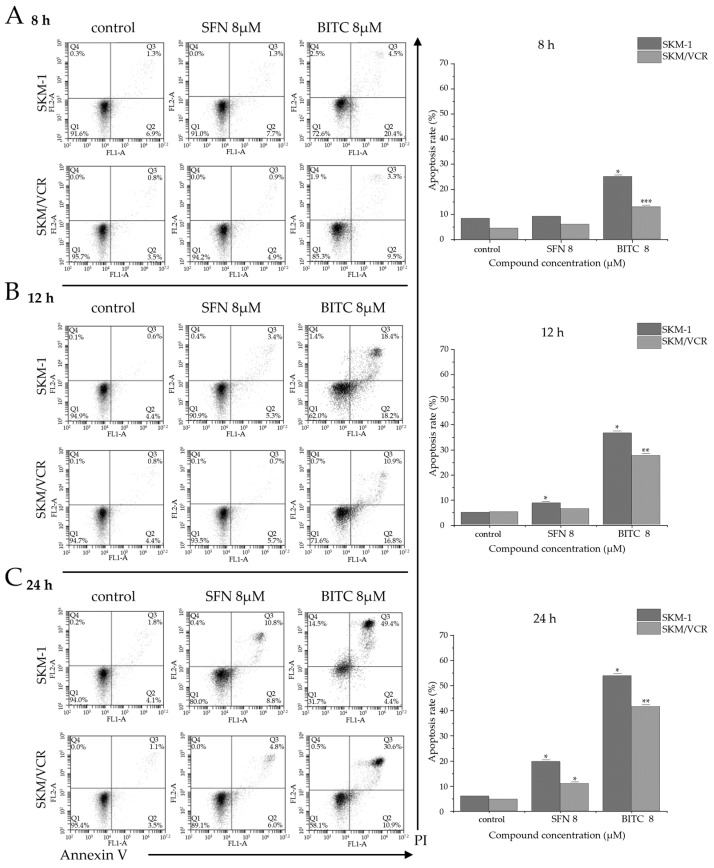
Apoptotic cell death induced by SFN and BITC in SKM-1 and SKM/VCR cell lines. Cells were treated with SFN or BITC (0 and 8 μM) for 8 h (**A**), 12 h (**B**) and 24 h (**C**). Apoptotic cell death was measured by annexin V/propidium iodide (PI) double staining by flow cytometric analysis as described in Materials and Methods. Each figure includes a representative cytogram (**left**) and quantitative data presentation (**right**). Significantly different from control at * *p* < 0.05, ** *p* < 0.01 and *** *p* < 0.001 as analyzed by *t*-test for 2-group comparisons.

**Figure 4 ijms-25-13511-f004:**
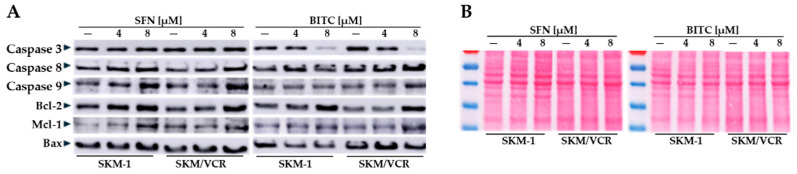
Effect of SFN and BITC treatment on the levels of Bcl-2 family proteins and caspases in SKM-1 and SKM/VCR cell lines. Cells were treated with SFN or BITC (0, 4 and 8 μM) for 12 h. (**A**) Total protein extracts (60 μg/line) were determined for SDS-PAGE gel electrophoresis and subjected to immunoblot analysis using anti-Bcl-2, anti-Bax, anti-Mcl-1, anti-caspase 3, 8, and 9 antibodies. Immunoblotting was performed at least three times for each protein using independently prepared lysates, and the results were comparable. (**B**) Ponceau S staining of total protein signals was used as an internal control. Densitometric quantification of the protein bands, which is fully consistent with the results described in the text, is documented in Appendix A.

**Figure 5 ijms-25-13511-f005:**
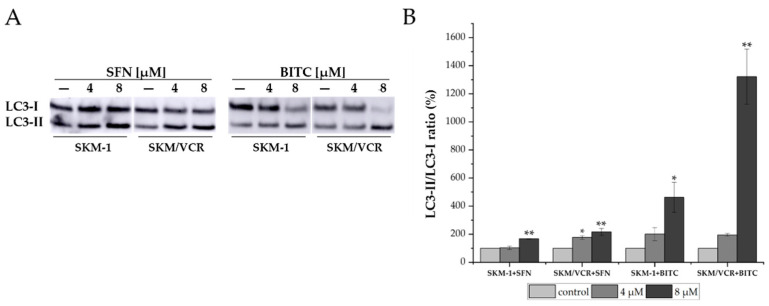
Western blot analysis of LC3-I and LC3-II. (**A**) Cells were treated with or without SFN or BITC (0, 4 and 8 μM) for 12 h and the ratio of LC3-II/I was detected using Western blotting. (**B**) LC3-II/LC3-I ratio after treatment with SFN and BITC in SKM-1 and SKM/VCR cell lines. Densitometric analysis of the immunoblot was performed by measuring the signal intensity of the bands using the National Institutes of Health ImageJ 1.52o program. Results are expressed as mean ± SD of 3 experiments vs. One-way ANOVA with Tukeyʹs post hoc test (* *p* < 0.05, ** *p* < 0.01 vs. control cells) was used to assess the significance of differences in the evaluated parameters.

**Figure 6 ijms-25-13511-f006:**
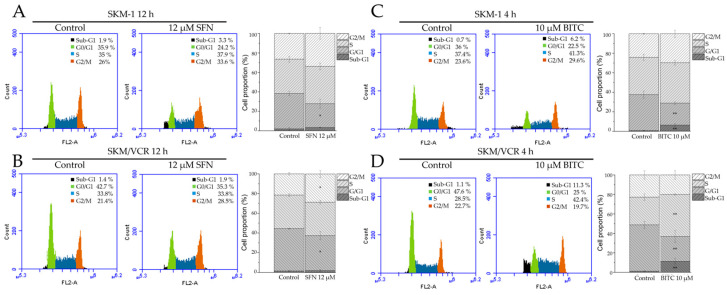
Effect of SFN or BITC on the cell cycle progression. Cell cycle distribution and quantification of the percentage of cells in each cell cycle phase of SKM-1 cells treated with 12 μM SFN for 12 h (**A**) and 10 μM BITC for 4 h (**C**) and SKM/VCR cells treated with 12 μM SFN for 12 h (**B**) and 10 μM BITC for 4 h (**D**). Values are expressed as the mean ± SD of three independent experiments. A *t*-test was used to compare the significance of differences between treated and control cells (* *p* < 0.05 and ** *p* < 0.01 vs. control cells).

**Figure 7 ijms-25-13511-f007:**
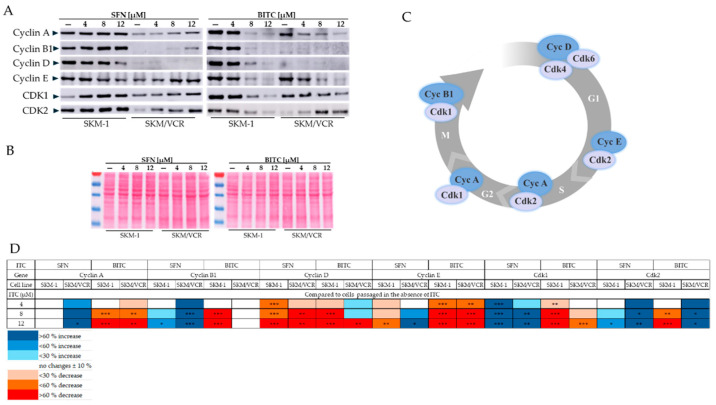
SFN and BITC affect the levels of different cell cycle regulators in human AML cell lines SKM-1 and SKM/VCR. (**A**) Cells were exposed to SFN or BITC (0, 4, 8, and 12 μM) for 12 h. Total protein extracts (60 μg/line) were determined for SDS-PAGE gel electrophoresis and subjected to immunoblot analysis using anti-cyclin A, B1, D, and E and anti-cyclin dependent kinases Cdk 1 and 2 antibodies. (**B**) Ponceau S staining of total protein signals was used as an internal control. (**C**) Cell cycle regulation. Cell cycle progression is controlled by a complex network of proteins, primarily cyclins and CDKs. CDKs are known to phosphorylate many proteins during the cell cycle to promote their progression. The catalytic activity of CDKs is tightly regulated by their association with specific cyclins, which accumulate during certain phases of the cell cycle. (**D**) Densitometric analysis of band densities was quantified using National Institutes of Health ImageJ 1.52o program. Densitometry was not performed on cyclin B1 immunoblots in SKM/VCR cells because the relative levels induced by BITC were very low or undetectable. Densitometric data are expressed as the mean ± SD of three independent experiments. One-way ANOVA with Tukey’s post hoc test (* *p* < 0.05, ** *p* < 0.01 and *** *p* < 0.001 vs. control cells) was used to assess the significance of differences in the evaluated parameters. The data are summarized in the table.

## Data Availability

Additional data and the resistant variants of SKM-1 cells are available from the authors upon reasonable request.

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
