# Peer review of "Sulforaphane and Benzyl Isothiocyanate Suppress Cell Proliferation and Trigger Cell Cycle Arrest, Autophagy, and Apoptosis in Human AML Cell Line"

_ijms, 2024, doi:10.3390/ijms252413511_

Round 1
Reviewer 1 Report
Comments and Suggestions for Authors
In the manuscript, the authors Bertova et al, have investigated the impact of isothiocynate (ITC) on AML. The authors have shown through western blot and flow based assays that BITC is a more potent at impacting SKM cells - both WT and variant types compared to SFN. Additionally, they point out that the mechanism of BITC is different in the SKM verus SKM/VCR cell lines. The authors have soundly carried out the experiments to prove their hypothesis. However, all the experiments are done using only one cell line - SKM. Is this phenotype seen in other AML cell lines as well, or is the mechanism specific to SKM cells? The authors should thus show that the same phenotype, at least for their main findings, is seen with some other AML cell lines at their disposal. Additionally, to improve the understanding of the readers, kindly provide an explanation of why specifically, BITC and SFN were used as ITCs to test among all the other ITCs available.
Author Response
Thanks a lot for your recommendation

Reviewer 2 Report
Comments and Suggestions for Authors
The study explores the effects of isothiocyanates (ITCs), natural sulfur-containing compounds, on acute myeloid leukemia (AML) cells. It examines the impact of sulforaphane (SFN, an aliphatic ITC) and benzyl isothiocyanate (BITC, an aromatic ITC) on cell proliferation and induction of cell death in the SKM-1 cell line and its multidrug-resistant (MDR) SKM/VCR variant, which exhibits P-glycoprotein overexpression. Both compounds were able to overcome MDR. Furthermore, the findings suggest that autophagy may act as a defense mechanism in response to ITC-induced apoptosis in AML cells.
I am enthusiastic about this study. Substances capable of overcoming MDR in AML are extremely valuable and deserve to be showcased. Even though their current structures are just at the beginning of the journey toward optimal pharmacological properties, future analogs could be developed based on the principles of rational drug design [1]. To enhance the quality of the presented work, the reviewer suggests the following:
Major point
- To better determine the kinetics of apoptosis, it would be beneficial to include at least two additional measurement time points.
- If the authors have results for other cell lines, it would be worthwhile to calculate selectivity SI indices to verify whether the mechanism is specific (or not) to leukaemia cells or general to cancer cells [2].
- It would be interesting to perform selected mechanistic studies as well as DNA content during cell cycle at concentrations higher than IC50 and with extended measurement times for cell proliferation e.g 48-72h However, this is not essential.
- Studies on the induction of increased metabolic activity would be excellently complemented by the use of flow cytometry or microscopic activity with JC-1 or MitoTracker dyes.
- The microscopy images confirming the autophagy process for SFN would be an excellent addition, although not a requirement.
- Adding resistance index (RI) data to the table in Fig. 2 would be a valuable enhancement [3].
Minor points:
- The Western blot presentation could be made clearer, for example, by labeling lanes with only minus (-) adding name of compounds.
- Combining Fig. 1 and Fig. 2 could improve the presentation of the data.
- Up to the point of the autophagy results, it would be safer to refer to SFN as a cytostatic agent. However, for BITC, based on the flow cytometry measurements, the term cytotoxic can already be appropriately used.
-
[1] Borowski, E., Bontemps-Gracz, M. M., & Piwkowska, A. (2005). Strategies for overcoming ABC-transporters-mediated multidrug resistance (MDR) of tumor cells. Acta biochimica Polonica, 52(3), 609–627.
[2] Lica, J. J., Wieczór, M., Grabe, G. J., Heldt, M., Jancz, M., Misiak, M., Gucwa, K., Brankiewicz, W., Maciejewska, N., Stupak, A., Bagiński, M., Rolka, K., Hellmann, A., & Składanowski, A. (2021). Effective Drug Concentration and Selectivity Depends on Fraction of Primitive Cells. International journal of molecular sciences, 22(9), 4931. https://doi.org/10.3390/ijms22094931
[3] Lica, J. J., Grabe, G. J., Heldt, M., Misiak, M., Bloch, P., Serocki, M., Switalska, M., Wietrzyk, J., Baginski, M., Hellmann, A., Borowski, E., & Skladanowski, A. (2018). Cell Density-Dependent Cytological Stage Profile and Its Application for a Screen of Cytostatic Agents Active Toward Leukemic Stem Cells. Stem cells and development, 27(7), 488–513. https://doi.org/10.1089/scd.2017.0245
Comments on the Quality of English LanguageThe English language is good and understandable. Minor corrections would be helpful.
Author Response
Thank you for your recommendation that we include to manuscript.
Reviewer 3 Report
Comments and Suggestions for Authors
The article is devoted to testing the effect of naturally occurring isothiocyanates (ITCs), sulforaphane (SFN) and benzyl isothiocyanate (BITC), on two acute myeloid leukemia (AML) cell lines, drug-sensitive SKM-1 and drug-resistant SKM/VCR variant, with special attention to cell proliferation, cell death, and drug resistance. In this article, the authors continue and extend their previous studies on the interactions of compounds of plant origin with cancer cells and the role of P-glycoprotein in drug resistance. First, they determined the cytotoxic effects of both compounds. BITC showed a greater ability to induce apoptosis than SFN, even in multidrug-resistant cells. At the cell cycle level, SFN primarily induces arrest in the G2/M phase, while BITC not only increases the sub-G1 population, which is indicative of apoptosis. The greater lipophilicity of BITC probably enhances its potency and contributes to its more pronounced apoptotic effect compared to SFN. Finally, the mechanism of ITC's impact on apoptosis, autophagy, and cell cycle was investigated based on protein markers using Western blot analysis.
The text is well-written and clearly explains the experiments performed. The approaches used are fairly straightforward and the experimental design is appropriate to the research objectives. Data are summarized and presented in figures, graphs, and tables to draw relevant conclusions. In the Discussion section, the results are broadly evaluated from several points of view. The authors conclude that BITC is a promising tool for targeting resistant AML.
There are a few issues that need to be pointed out:
Questions:
- Given the reactivity of SFN and BITC, are the concentrations used realistic regarding cell exposure? It can be expected that these compounds will readily react with the abundant proteins present in a medium containing 12% fetal bovine serum.
- Are ITCs somehow specific to have a preferential effect on the cancer cells, or do they have a similar effect on normal cells of the same origin
Minor issues:
- missing blot description (Figure 5A)
- “any of these three caspases” (Line 355) – based on the context, “two caspases 8 and 9” are probably meant; caspase 3 is discussed much later (Line 370)
In conclusion, the paper is well written and, with minor revisions, is fully acceptable for publication in IJMS.
Author Response
Thanks a lot for your recommendation.

Round 2
Reviewer 2 Report
Comments and Suggestions for Authors
While the Authors have made progress in addressing the initial comments, the manuscript still requires some corrections before it is ready for publication.
A detailed description of the remaining issues and suggested improvements has been provided in my response to the authors' comments. These revisions focus primarily on enhancing data presentation, clarifying certain aspects of the methodology, and improving the overall readability of the figures and text.
I appreciate the Authors’ efforts thus far and hope these final adjustments will ensure the manuscript meets the journal's high standards.
Kind regards

The English language is good-written.
Author Response
Thank you for valuable recommendation.

Round 3
Reviewer 2 Report
Comments and Suggestions for Authors
Thank you for implementing the changes. I have noticed a minor point that could be addressed before publication. In Fig. 3A, SKM/VCR - BITC 8 μM, the discriminator is slightly shifted to the right compared to the rest (attached file). It would be good if this could be corrected prior to publication to avoid disrupting the overall impression of this work. I congratulate the authors on their work and wish them success in continuing their research.

The English language is well-written.
Author Response
Thank you for recommendation.
We replaced Figure 3 with the corrected version.